# Poly(β-cyclodextrin)-Activated Carbon Gel Composites for Removal of Pesticides from Water

**DOI:** 10.3390/molecules26051426

**Published:** 2021-03-06

**Authors:** Gianluca Utzeri, Luis Verissimo, Dina Murtinho, Alberto A. C. C. Pais, F. Xavier Perrin, Fabio Ziarelli, Tanta-Verona Iordache, Andrei Sarbu, Artur J. M. Valente

**Affiliations:** 1Coimbra Chemistry Centre, Department of Chemistry, University of Coimbra, 3004-535 Coimbra, Portugal; uc2015276036@student.uc.pt (G.U.); luisve@gmail.com (L.V.); dmurtinho@ci.uc.pt (D.M.); pais@ci.uc.pt (A.A.C.C.P.); 2Laboratoire MAPIEM, Université de Toulon, 83041 Toulon CEDEX 9, France; francois-xavier.perrin@univ-tln.fr; 3CNRS, Centrale Marseille, FSCM, Aix Marseille University, 13397 Marseille CEDEX 20, France; fabio.ziarelli@univ-amu.fr; 4National Institute for Research & Development in Chemistry and Petrochemistry-ICECHIM, Splaiul Independenței 202, 060021 București, Romania; iordachev.icechim@gmail.com (T.-V.I.); andr.sarbu@gmail.com (A.S.)

**Keywords:** β-cyclodextrin, activated carbon, hydrogels, pesticides, remediation

## Abstract

Pesticides are widely used in agriculture to increase and protect crop production. A substantial percentage of the active substances applied is retained in the soil or flows into water courses, constituting a very relevant environmental problem. There are several methods for the removal of pesticides from soils and water; however, their efficiency is still a challenge. An alternative to current methods relies on the use of effective adsorbents in removing pesticides which are, simultaneously, capable of releasing pesticides into the soil when needed. This reduces costs related to their application and waste treatments and, thus, overall environmental costs. In this paper, we describe the synthesis and preparation of activated carbon-containing poly(β-cyclodextrin) composites. The composites were characterized by different techniques and their ability to absorb pesticides was assessed by using two active substances: cymoxanil and imidacloprid. Composites with 5 and 10 wt% of activated carbon showed very good stability, high removal efficiencies (>75%) and pesticide sorption capacity up to ca. 50 mg g^−1^. The effect of additives (NaCl and urea) was also evaluated. The composites were able to release around 30% of the initial sorbed amount of pesticide without losing the capacity to keep the maximum removal efficiency in sorption/desorption cycles.

## 1. Introduction

One of the major challenges facing society nowadays is the development of effective methodologies to eliminate or reduce air, soil and water (surface and groundwater) contamination as a consequence of growing anthropogenic activities. Urbanization and industrialization has led to a massive release of different contaminant classes such as heavy metal ions [1], volatile organic compounds (VOCs) [2], polyfluoroalkyl substances (PFAs) [3], persistent organic pollutants (POPs) [4] and pesticides [5] into the environment. These xenobiotics can be harmful for both human health and ecosystems, causing sizable changes in the flora and fauna equilibria. Depending on their physical–chemical properties, they show different toxicity levels and mechanisms of action. Pesticides, which include biocides and plant protection products (PPPs), are mainly applied in the agriculture for improving harvests. Due to soil persistence, water contamination and food chain concentration, pesticides can affect directly or indirectly a wide range of organisms [6]. It is consensual that pesticides can cause adverse effects to human health leading to acute or chronic disorders, with effects being a function of time and amount of exposure [7,8]. Ideally, these products should be highly specific and, thus, selectively toxic to the target organisms. However, most of the pesticides have a broad spectrum of action and only ca. 0.1% reach the intended target [9]. Due to the agrochemicals’ highly negative impacts, its removal from soil and water is a major challenge [10,11]. Hence, the improvement of environmental remediation techniques [12], e.g., biological treatment [13] and advanced oxidation processes [14], is pivotal. Sorption processes and materials have wakened great attention being straightforward in application and often reusable, showing good efficiency and advantageous cost effectiveness. A wide range of adsorbent materials, including clays [15], alumina, mesoporous metal oxide [16,17,18] and zeolites [19,20,21], can be used for an extensive range of applications [22], albeit, activated carbon (AC) remains a most prominent material for environmental contaminants removal due to its high sorption efficiency. This includes the effective sorption of different pesticides as summarized in Table 1.

Despite that, two major AC drawbacks can be pointed out: the high production cost and the low regeneration capacity [44]. Low-cost materials are vital as AC alternatives and a wide range of sorbents have gained increasing attention. Amidst them, hydrogels—materials formed by 3D hydrophilic polymer network capable of sorption large amounts of water—have been applied to environmental remediation processes [45,46]. Among these polymers, natural polysaccharides such as starch, chitosan and pectin have been increasingly used as adsorbents due to being biodegradable, inexpensive, non-toxic, abundant, and renewable [47,48]. The presence of reactive chemical groups such as amine, hydroxyl or carboxyl permit easy modification as well as notable selective interactions. For these reasons, polysaccharides and their derivatives are smart alternative sorbent materials with remarkable sorption efficiency and might as well be easily regenerated by desorption cycles. Cyclodextrins (CDs) are cyclic oligosaccharides with outstanding properties, including the ability to form host-guest supramolecular complexes [49,50,51]. Consequently, the use of CDs arises as a natural choice for preparing sorbent materials [52,53,54] as they can be used as monomers for hydrogel formation [55,56,57] and, simultaneously, provide a significant number of active sites for interaction with less water soluble pollutants as, e.g., pesticides [58]. The synthesis of CD-based hydrogels relies on a covalent binding of a CD to a solid matrix or a direct linkage of CDs by using a crosslinker (epichlorohydrin and diisocyanates are commonly used [59,60,61]).

Additionally, we also want to address the question: is it feasible preparing sorbent material with the capability to provide for active substances to the soil, when it is needed, without further maintenance? For such purpose, the sorbent material should be able to show high sorption-desorption cyclic performance.

In this paper, we describe the synthesis and characterization of poly(β-cyclodextrin)s (PCD) and activated carbon composites, at different weight ratios. The PCD hydrogels were initially prepared by copolymerization of β-CD with hexamethylene diisocyanate (1,6-HDI), acting as crosslinker, through a carbamate forming reaction. The activated carbon has been used due to its high sorption performance for a vast type of pollutants, whilst the PCD was chosen due to its low swelling degree, allowing for a long-term encapsulation of the AC and concomitantly keeping high permeation rates. The obtained hydrogel composites were assessed towards the removal of two active agrochemical substances: cymoxanil (CYM) and imidacloprid (IMD), a fungicide of the cyanoacetamide oximes class and an insecticide of the neonicotinoid class, respectively (Scheme 1). The evaluation was made on the basis of a thorough study on sorption kinetics and isotherms as well as on the structural characterization of composites before and after sorption. Besides, the effect of different additives (NaCl and urea) and sorption/desorption cycles on the gel performance was also evaluated.

## 2. Results and Discussion

### 2.1. Poly(β-cyclodextrin) Synthesis and Characterization

Crosslinked PCD was prepared by reaction of the primary hydroxyl groups of β-CD with the isocyanate groups of 1,6-HDI (Figure 1a), forming a urethane linkage. The reaction is catalyzed by BTL, which acts as Lewis acid by coordination of the Sn with the nitrogen atom of the isocyanate and, thus, facilitates the attack of the oxygen of the hydroxyl group to the carbonyl of the isocyanate.

In the PCD/AC composites, the AC is retained by the PCD crosslinked matrix (Figure 1b,c).

A thermogravimetric characterization has been performed to assess the effect of AC addition at 5% and 10% (*w*/*w*) on the thermal stability of poly(*β*-CD). The TGA and dTG curves of adsorbent materials are shown in Figure 2a,b. All adsorbents showed an initial loss of mass up to ca. 12% with a maximum temperature around 100 °C, associated with solvent evaporation. PCD showed a typical second degradation step between 250 °C and 350 °C with a maximum degradation rate at 277 °C and weight loss of ca. 50%; such thermal event can be assigned to the *β*-CD degradation [62,63]. All poly(β-CD)-based materials revealed a further thermal degradation at around 452 °C. That can be justified by the cleavage of the ester bond between *β*-CD and 1,6-HDI [64]. It should be stressed that the latter can also be indicative of a successful synthesis of PCD. Taking into account just these two thermal changes, it can also be noticed that the corresponding maximum degradation temperatures, for PCD/AC_10%_, show a slightly decrease; i.e., they occur at 262 °C and 449 °C, suggesting that the incorporation of a higher content of AC has a negligible effect on the stability of the poly(β-cyclodextrin). We can hypothesize that an increase in the AC content might decrease the intermolecular H-bonds interactions between CDs affecting its thermal stability. An interesting process is also observed as an additional thermal transition for PCD/AC_5%_ and PCD/AC_10%_ at 322 °C and 327 °C. These transitions seem to indicate that the incorporation of AC does not affect equally the polymeric structure; i.e., the hypothetical (microscopically) heterogeneous density of AC might induce polymeric zones of higher and lower stability. This is supported by the weight loss percentage for these two regions being similar to that observed just for PCD. Finally, the decrease in the mass loss shown in the TG of AC can be assigned to devolatilization [65].

The synthesis of PCD and its composites was also analyzed by FT-IR. The corresponding spectra are shown in Figure 2c. Considering the FTIR spectrum of the PCD it is possible to observe a broad band around 3400 cm^−1^ due to the asymmetrical and symmetrical stretching vibration of the -OH groups. The two bands at 2935 cm^−1^ and 2862 cm^−1^ are associated to the methylene groups stretching vibrations of the 1,6-HDI crosslinker. The absence of a peak at 2270 cm^−1^, corresponding to the isocyanate group, indicates a complete polymerization reaction [64]. A characteristic band appears at 1710 cm^−1^, which is due to the stretching vibration of the carbonyl group of the ester bond between the -OH at C-6 position of the *β*-CD and the terminal carbonyl of the isocyanate group of 1,6-HDI. The presence of the amide groups is also confirmed by the band at 1547 cm^−1^ relative to the bending vibration of the -NH group in secondary amide. The band at 1265 cm^−1^ results from the overlapping of bands due to the interaction of the -NH bending vibration and -CN stretching vibration of the amide groups. The band at 850 cm^−1^ is characteristic of the α - 1 → 4 glycosidic bond [66]. Finally, the weak bands between 800 and 600 cm^−1^ can be assigned to the -NH wagging vibrations of the secondary amide groups [67].

Comparing the spectra of neat-PCD and its composites with activated carbon at 5% and 10%, it is not possible to observe any characteristic bands that can confirm or reveal the presence of the activated carbon, probably due to the weak adsorption bands of the AC. However, in a more detailed analysis, it is possible to observe two weak peaks at the high frequency region around 3700–3600 cm^−1^, indicating the presence of hydroxyl groups. The bands between 1500 and 1000 cm^−1^ can be assigned to C-O stretching and -OH bending vibrations. The bands in the region 950–800 cm^−1^ are probably associated to the C-H vibration [68]. However, from the ^13^C CPMAS solid-state NMR spectra (Figure 2d), which compares the spectrum of PCD with that of PCD/AC_10%_, it is observed a broad and low carbon resonance, around 125.5 ppm, associated with aromatic structures of AC moiety [69]. The other resonances can be assigned to four carbon resonances of the glucopiranose unit: C1 (101.1 ppm), C4 (82.1 ppm), C2/C3/C5 (72.7 ppm) and C6 (60.7 ppm) [70]. Moreover, it can be noticed at low field a carbon resonance C7 (157.3 ppm) associated to the carbonyl in the carbamate (urethane) group while at high field two carbon resonances—C8 (41.6 ppm) and C9 (27.7ppm)—related to the carbon of the methylene groups in the aliphatic chain of 1,6-HDI, can be observed.

The swelling behavior of different gels were also evaluated by the analysis of the water uptake capacity (Table 2). It can be seen that the swelling degree for the PCD is not high, in line with a semi-rigid (not soft) gel when handled, and the encapsulation of AC leads to a decrease in the *SW*. This might be related with the structural effect produced by the AC into CD structure as seen by the thermogravimetric and BET analysis. Concerning the latter, it can be observed that the incorporation of AC into pBC increases by 3 times the surface area of the composite. This increase is accompanied by a continuously decrease in the average pore diameter with an increase in the AC percentage.

### 2.2. Sorption Kinetics

The adsorption kinetic analysis provides valuable information about the adsorption efficiency of an adsorbent material and a better understanding of the adsorption mechanism. It should be stressed that these experiments were carried out at two initial pesticide concentrations 200 ppm and 500 ppm.

Figure 3 shows representative sorption kinetic data for IMD and CYM onto PCD/AC_10%_. In general, the maximum sorption capacity is attained (taking a cumulative sorption capacity of 80%) after ca. 10 and 20 h. Such slow sorption rate might be related to the relatively low swelling degree of these gels. In fact, independently of the model [71], there is an inverse relationship between the diffusion coefficient (or rate constant) and the polymer volume fraction and thus with the swelling degree; i.e., the fractional drag action on adsorbate movement inside the gels increase by decreasing the swelling degree, and consequently, their kinetic rate constant [72]. It is also worth noticing that the time needed to reach the sorption equilibrium is also high and similar in magnitude to that found for PCD-based hydrogels.

Additionally, the kinetic data were evaluated by the PFO and PSO (Equations (7) and (8)); the fitting parameters for all sorption systems are summarized in Table 3 and Table 4 and are represented by solid and dashed lines in Figure 3. Generally, the determination coefficients (*R^2^*) values are higher than 0.95, indicating a good fit. However, once the *R*^2^ values are similar for the fitting of both equations to the experimental data, the AIC are used to conclude on the best fitting. It can be concluded that for all systems the sorption kinetic follows a pseudo-second order mechanism. It is worth noticing that the *q*_e_ values obtained by Equation (8), representing the amount of pesticide sorbed at equilibrium time, denote that the constancy of *q_e_* value is not completely reached after 4 days. Hence, a chemisorption nature is suggested which can be justified by the occurrence of hydrogen bonds or inclusion complexes formation. That hypothesis can also explain the higher adsorbed amount of pesticide at higher concentration and larger contact time as observed in the experimental design and kinetic and equilibrium analysis. Another issue that deserves a comment is the effect of the initial concentration on the *k*_2_ rate constants: by increasing the initial concentration from 200 ppm to 500 ppm, *k*_2_ values decrease significantly. This indicates that other phenomena than sorbent-sorbate interaction occur; an example of that might be an increase of sorbate-sorbate interactions upon concentration increase [46]. Both these findings are in agreement with the sorption isotherms as will be discussed in the next section.

### 2.3. Sorption Isotherms

Sorption isotherms for IMD and CYM onto gel composites and activated carbon have been measured and are shown in Figure 4 and Figure 5, respectively. It can be seen that the sorption mechanism changes with the sorbent. In fact, for PCD the sorption isotherms are almost linear, following a quasi-Henry model, suggesting a diffusion of pesticides into the gel water free volume or a weak interaction between the pesticides and the cyclodextrins. By increasing the incorporation of AC into PCD the dependence of the *q_e_* as function of *C_e_* becomes more concave and the sorption becomes clearly favorable. For the activated carbon, the sorption follows an s-shape isotherm, with significantly low *C*_e_ values, suggesting an initial highly favorable sorption towards AC. Such behavior is independent on the IMD or CYM. A deeper analysis has been done by fitting the Freundlich and Sips equations to the experimental data (see fitting lines in Figure 4 and Figure 5); the corresponding fitting parameters are gathered in Table 5 and Table 6. In general, the Sips model fits better the experimental sorption data, considering the determination coefficient and the AIC data. From the analysis of the fitting parameters reported in Table 5 and Table 6, it can be noticed that the maximum sorption capacity increases by increasing the amount of AC into the gel matrix. It can also be observed a similar stability constant, for IMD and CYM, registered for PCD and PCD/AC_5%_; however, for the PCD/AC_10%_ and AC the interaction between IMD and the adsorbents is significantly higher. This might be justified by the occurrence in higher extent of London dispersion forces between the activated carbon and the more hydrophobic pesticide, IMD.

The interaction between IMD and CYM and the four different adsorbents was complementary evaluated by scanning electron microscopy (Figure 6).

The micrographs show an irregular and amorphous surface for PCD and PCD/AC_5%_ (Figure 6a,b—top row) and become smoother by increasing the amount of AC. After 24h in contact with the aqueous solution of CYM and IMD, respectively, significant morphological alterations have been observed. Thus, the surface morphology of PCD becomes significantly more featureless in the presence of IMD and a similar behavior occurs for PCD/AC_5%_, showing that for this gel composite, the cyclodextrin effect is predominant. On the other hand, an opposite effect is observed for PCD/AC_10%_ and AC; i.e., the sorption of CYM by PCD/AC_10%_ (c)-middle row) leads to a highly jagged surface, which is enhanced in the presence of IMD. In the case of AC the effect of both pesticides on the adsorbent surface morphology do not follow the same trend, although they are different when compared with the pristine adsorbent. The former evidence is in close agreement with the fact that IMD interacts more strongly with the gel than the CYM thus producing a more significant surface modification; the latter might be related with the high performance of the activated carbon for the sorption of both pesticides reaching a removal efficiency higher than 98% (Figure 7) and a maximum amount of sorbed CYM and IMD of 53.4(±0.7) and 49.5(±0.3) mg g^−1^ (Appendix A) respectively. The analysis of Figure 7 and Appendix A deserves further comments. The performance of PCD is the poorest of all adsorbents (tested with *RE* lower than 35% and 20% for IMD and CYM, respectively), in agreement with the higher affinity of the aromatic towards the hydrophobic cavity of the CD. However, for the composite gel adsorbents, the removal efficiency increases by increasing the amount of encapsulated AC and, as a general trend, decreases by increasing the initial concentration of pesticide. Such a behavior might be linked to the tendency of pesticides to form dimers [73], which might hamper the sorption process—as previously discussed. It should be noticed, however, that there is an increase in the *q_e_* values with an increase in the initial concentration of pesticides (Appendix A). The maximum removal efficiencies of CYM by PCD/AC_5%_ and PCD/AC_10%_ are 45(±2) and 73(±2)%, respectively, whilst for IMD are 55.9(±0.6) and 80.9(±0.7)%, respectively. Such behavior is remarkable for two different reasons: the removal efficiency is higher for the lowest initial concentration (i.e., 50 ppm) and the use of lower amounts of AC leads to a significant level of removal. Looking at the *q_e_* values, an increase in the amount of AC from 5% to 10% doubles and triples the *q_e_* values for CYM, compared to PCD; again, the AC is playing a main role on the sorption efficiency.

For the sake of comparison, the values of *RE* and *q*_m_ reported in the Table 1 for the removal of IMD by using AC [40,41] are lower than those obtained by us, and are similar to those corresponding to the use of PCD/AC_5%_ and PCD/AC_10%_.

### 2.4. Effect of Additives on the Pesticide Sorption Isotherms

Saline soils are composed of several types of dissolved salts but NaCl is the most prevalent one, which is why the effect of salinity on the sorption of pesticides is of great importance [74]. Furthermore, urea is a nutrient often used in cropping together with different types of pesticides and has proved to be suitable as a carrier for some of the herbicides [75]. Hence, the influence of additives like NaCl and urea has been assessed in the attempt to evaluate the sorption efficiency of composite materials in simulated environments. The effect of ionic strength and additives on the sorption isotherms of CYM and IMD have been evaluated by using NaCl (1 g L^−1^) and urea (1 g L^−1^) as model molecules and data are shown in Appendix A, respectively. Generally, it can be observed an analogous behavior of the adsorbent materials for IMD in presence of urea and NaCl, with favorable and cooperative adsorption process as it is suggested from the value of *n* and *K_s_* parameters (Appendix A).

However, there are some issues that deserve further discussion. The amount of sorbed cymoxanil and the corresponding removal efficiencies for cymoxanil onto PCD and PCD/AC_5%_, in the presence of NaCl, are higher than in its absence. Namely, the *RE* is twice higher than in the NaCl absence. The structure of cymoxanil shows a significant number of amine groups able to interact with primary and secondary hydroxyl groups of β-CD by H-bonding. Simultaneously, it is known that intermolecular CD interactions might occur strongly [76]. Once this effect is more significant for the gels with lower amount of AC, we may hypothesize that the screening effect [77] caused by an increase in the ionic strength decreases the number of CD-CD interactions (in spite of its short-range nature) [78,79], leaving the CD available to interact with CYM. The sorption enhancement obtained for PCD/AC_5%_ might also be related to a more porous surface morphology. Even so, and although urea can be removed by activated carbon [80], it is worth noting that no modification of the pesticide *RE* is observed. In the case of IMD removal no significant changes are observed in the presence of either NaCl or urea.

### 2.5. Reusability Assessment

Three cycles of adsorption/desorption have been performed using ca. 0.1 g of each adsorbent in 10 mL of 500 ppm solution of IMD and CYM, respectively. The results are shown in Figure 8. In general, it can be noticed a slightly increase of the sorption capability along the three cycles of sorption/desorption for both pesticides. The average values of *q_e_* and *RE%* for IMD sorption onto PCD, PCD/AC_5%_ and PCD/AC_10%_ agree with the experimental values obtained by equilibrium analysis. That behavior is more pronounced for CYM where it can be also observed a decrease of the desorption rate for the three adsorbent materials. The same result is achieved with PCD/AC_10%_ for IMD while PCD and PCD/AC_5%_ reach the equilibrium at the first cycle both in sorption and desorption steps. Hence, it can be suggested that the presence of sorbate molecules plays a prominent role promoting the sorption process via cooperative mechanism due to sorbate-sorbate interaction. The results also suggest that the sorption mechanism occurs via dual physi- and chemisorption.

In addition, AC does not show any ability to desorb either of these pesticides by diffusion (data not shown). Thus, we can conclude that the desorbed capacity of the composite gels is mainly due to the PCD. Besides this, it is interesting to observe from the data in Appendix A that the presence of AC on the composite significantly ruled the IMD release in contrast with CYM release probably determined by concentration gradient.

## 3. Materials and Methods

### 3.1. Materials

*β*-cyclodextrin (*β*-CD) was purchased from ACROS-Organics (MW = 1134.98 g·mol^−1^, 98%, CAS 7585–39–9). For the poly(*β*-cyclodextrin) synthesis, hexamethylene diisocyanate (MW = 168.19 g·mol^−1^, purity ≥ 98%, CAS 822–06–0, 1,6-HDI) and dibutyltin dilaurate (MW = 631.56 g·mol^−1^, 95%, CAS 77–58–7, BTL) were supplied by Sigma-Aldrich (Steinhem, Germany). The *N,N*-dimethylformamide (DMF) used in the syntheses was dried by treatment with CaO (previously activated at 500 °C) for 24 h, followed by decanting and stirring with NaOH (solid) for 1 h. After further decantation, the solvent is distilled and stored over 4 Å molecular sieves. Activated carbon (AC, granular, pure) was purchased from Riedel-de Haën (Seelze, Germany). Cymoxanil (MW = 198.18 g·mol^−1^, 98.9%, CAS 57966–95–7, CYM) and imidacloprid (MW = 255.66 g mol^−1^, 100%, CAS 138261–41–3, IMD) were supplied by Sigma-Aldrich (Steinhem, Germany).

The other chemicals were used without further purification process. All solutions were prepared with Millipore-Q water obtained with gradient water purification system *Direct*-*Q^®^* 3UV-R.

### 3.2. Synthesis of Poly(β-cyclodextrin) and Its Composites

Poly(*β*-cyclodextrin) was synthesized following a previous work reported in literature [81]. Briefly, 1.8 mmol of *β*-CD was solubilized in anhydrous DMF, under stirring and in a nitrogen atmosphere. After complete dissolution, one drop of BTL was added, as catalyst. Then, a DMF solution of 1,6-HDI (7.2 mmol) as crosslinker was added dropwise to the mixture and heated at 70 °C for 24 h. Likewise, two PCD composites (PCD/AC) were prepared by encapsulating 5% and 10% (*w*/*w*) of activated carbon. The size of AC powder has been measured by sieving and the following size distribution was obtained: 300–212 µm: 49%; 212–180 µm: 7%; 180–75 µm: 3%; 75–38 µm: 17%; 38–25 µm: 13%; and <25 µm: 9%.

The obtained polymers were washed first with an excess of chloroform, under stirring for 12 h at room temperature, to remove the organic solvent and then, with ultrapure water, under stirring at room temperature, to remove the unreacted *β*-CD [81].

Finally, the solids were collected by filtration in filter paper and dried in an oven at 60 °C. Afterwards, a second drying step was performed at 160 °C by using a thermo-balance (A&D MS-70, Tokyo, Japan) until reaching a mass variation ≤0.001% min^−1^, ensuring a complete solvent removal. The xerogels of PCD and PCD/ACs appear as macroscopically homogeneous solids, white and dark grey, respectively. The adsorbents were used as powder. Obtained percentage yields (weight): PCD = 89.3%; PCD-AC_5%_ = 92.5%; PCD-AC_10%_ = 78.0%.

### 3.3. Physico–Chemical Characterization of Absorbent Materials

The chemical structure of neat-PCD, AC and their composites was evaluated by Fourier transform infrared spectroscopy (FT-IR) (Nicolet 6700 FT-IR, Thermo Scientific, Waltham, MA, USA) in a wavelength interval of 4000 and 400 cm^−1^ using KBr pellets as reference. Pellets were prepared at 1% (*w*/*w*) for PCD and PCD/AC_5%_ and at 0.5% (*w*/*w*) for PCD/AC_10%_ and AC with KBr.

The incorporation of AC into PCD was further assessed by solid-state nuclear magnetic resonance (NMR) using ^13^C cross polarization magic angle spinning (CPMAS) technique.

The ^13^C CPMAS solid state NMR spectra were obtained on a Bruker Avance HD-400 MHz NMR spectrometer operating at a ^13^C resonance frequency of 106 MHz and using a commercial Bruker double-channel probe. To obtain a good signal-to-noise ratio in ^13^C CPMAS experiment 20 K scans were accumulated using a delay of 2.5 s. The ^13^C chemical shifts were referenced to tetramethylsilane and calibrated with glycine carbonyl signal, set at 176.5 ppm.

Thermogravimetric (TGA) and differential thermogravimetric (dTG) analysis of PCD, PCD/AC (5% and 10%) and AC in xerogel forms have been carried out by using TG209-F3 Tarsus (Netzsch Instruments, Selb, Germany). The analysis was performed in TiO_2_ plates using approximately 4–5 mg of adsorbent, in the 25 to 800 °C temperature range, at a heating rate of 10 °C·min^−1^, under nitrogen atmosphere with a flow rate of 50 mL·min^−1^.

The surface morphology of the xerogel was characterized by scanning electron microscopy (SEM) (Gemini 2-Zeiss, Merlin-Zeiss, Oberkochen, Germany,) at 2.00 kV with magnification 10,000×. The samples were previously frozen using liquid nitrogen followed by freeze drying (Labconco, Kansas City, MO, USA, model Free-Zone 4.5) before being sputter-coated with a thin gold layer. PCD, PCD/AC_5%_, PCD/AC_10%_ and AC were analyzed before and after 24 h of contact with 200 mg L^–1^ IMD and CYM aqueous solutions, respectively, to determine the morphological effects.

The Brunauer Emmett Teller (BET) specific surface area was obtained through nitrogen adsorption (ASAP 2000, Micrometrics, Norcross, GA, USA).

The water uptake capacity was determined by swelling analysis, performed in triplicate, using ca. 50 mg of adsorbent at 25 °C. The swelling degree (*SW*) was calculated by Equation (1):(1)$$SW\%\;=\frac{\left(m_{e}-m_{0}\right)}{m_{0}}\times 100$$
where *m*_0_ and *m_e_* (g) are the masses of the adsorbent as xerogel and swollen state at equilibrium, respectively.

### 3.4. Sorption Analysis

The optimal experimental conditions for the equilibrium and kinetic sorption analysis were assessed by performing a 2^3^ factorial analysis. The effect of three parameters: volume (5 and 10 mL), amount of adsorbent (50 and 100 mg) and concentration of CYM (7.5 and 22.5 ppm) and IMD (5.0 and 15 ppm) was considered. The concentration values were decided taking the analytical range into account. The linearity of the analytical methods, the analytical range, limit of detection (LOD) and limit of quantification (LOQ) were statistically determined. The quantification of CYM and IMD concentrations were obtained by UV-Vis spectroscopy (Shimadzu UV-2450, Kyoto, Japan) between 200 nm and 500 nm, at the maximum wavelengths: 242 nm and 270 nm, respectively. The statistical parameters supporting the normal data distribution, absence of outliers as well as the linearity of the calibration data are reported in Appendix A, respectively. UV-Vis spectra and calibration curve are shown in Appendix A while the analytical parameters of IMD and CYM are reported in Appendix A.

Sorption analysis of CYM or IMD were performed in aqueous solutions at 25 °C and shaken at 120 rpm in an incubator LABWIT (ZWI-100H, Shangai, China) using a weighted mass of neat–PCD, PCD/AC_5%_, PCD/AC_10%_ and pristine AC. Nylon teabags (100-mesh nylon screen) were used to ensure a complete immersion of the adsorbents in the bulk phase. In the concentration range at which pesticide sorption was evaluated, control experiments were carried out showing that no measurable sorption of pesticides onto teabags has been detected. The efficiency of removal (*RE%*) of both active ingredients was calculated by using Equation (2),
(2)$$RE\%=\;\frac{\left(C_{0}-C_{e}\right)}{C_{0}}\times 100$$
where, *C*_0_ and *C_e_* (mg L^−1^) are initial and equilibrium concentration in the analytical samples, respectively. The amount of CYM or IMD adsorbed was computed by using Equation (3)
(3)$$q_{e}=\frac{\left(C_{0}-C_{e}\right)}{m}\times V$$
where, *q_e_* (mg g^−1^) is the amount of active ingredient adsorbed per gram of adsorbent at equilibrium state, *V* (L) is the volume of the analytical solutions and *m* (g) is the mass of the adsorbent. Sorption isotherms were performed by using a concentration range from 0 to 500 mg L^−1^ for CYM and IMD. Additionally, sorption isotherms in a pesticide aqueous solution, containing either NaCl or urea (both at 1.0 g L^−1^) were also performed. A mass of adsorbate of ca. 0.1 g was used.

Based on the experimental data, and assuming a heterogeneous adsorbent, two isotherm models have been chosen: Freundlich [82] and Sips [83,84] (Equations (4) and (5)), respectively,
(4)qe=KFCe1n
(5)qe=qmKsCe1ns(1+KsCe1ns)
where, *K_F_* (mg g^−1^ L^1/n^ mol^−1/n^) and *K_S_* (L^1/n^ mol^−1/n^) are the Freundlich and Sips constants, *q_m_* (mg g^−1^) is the maximum sorption capacity per unit weight of the adsorbent and 1/*n* and 1/*n*_S_ are heterogeneous related factors. Equation (5) predicts a monolayer sorption, characteristic of the Langmuir equation [85], to higher adsorbate concentrations or when *n*_S_ equals to 1. The goodness of the fitting was assessed by the determination coefficient (*R*^2^) and the Akaike information criterion (*AIC*)—Equation (6) [86]:(6)$$AIC=Nln\frac{SS}{N}+2k$$
where, *N* is the number of experimental points, *SS* is the sum of squared errors and *k* is related to the number of fitted parameters.

Sorption kinetic analysis were performed using ca. 0.1 g of PCD, PCD/AC_5%_, PCD/AC_10%_ and AC in 10 mL of IMD and CYM aqueous solutions at 200 ppm and 500 ppm. The sorption kinetic mechanism was assessed by using the pseudo–first order (PFO) and pseudo–second order (PSO) kinetic models (Equations (7) and (8)), respectively.
(7)$$q_{t}=q_{e}\left(1-e^{-k_{1}t}\right)$$
(8)$$q_{t}=\frac{k_{2}q_{e}^{2}t}{1+q_{e}k_{2}t}$$
where, *q_t_* (mg g^−1^) is the amount of adsorbate at defined interval of times *t* (min). *k_1_* (min^−1^) and *k*_2_ (g mg^−1^ min^−1^) are the rate constants for PFO and PSO, respectively.

Sorption kinetics and isotherms were performed in triplicate.

### 3.5. Reusability Assessment

The reusability of the adsorbent materials was evaluated by performing three sorption/desorption cycles, using experimental conditions similar to those used for sorption kinetics. Desorption capability (*DC%*) was computed by using Equation (9)
(9)$$DC\%=\frac{C_{e_{des}}}{C_{e_{sor}}}\times 100$$
where, *Ce_sor_* and *Ce_des_* (mg·L^−1^) are the equilibrium concentrations of the sorbed and desorbed pesticide, respectively. The analysis was performed in triplicate.

## 4. Conclusions

Poly(β-cyclodextrin)-activated carbon composite materials were synthesized and their performance for the removal of cymoxanil and imidacloprid—pesticides highly used in agriculture of the South of Europe—was evaluated. The use of poly(β-cyclodextrin) and activated carbon allows to use a highly efficient material (AC) with an amphiphilic material (CD) with high capacity to interact with low molecular weight compounds. Additionally, the use of PCD enables to obtain a hydrogel with a low swelling degree. This feature seems to be of utmost importance because it prevents the release of AC and simultaneously allows the partial release of pesticides.

Cyclodextrin was polymerized by using 1,6-HDI as crosslinker; two composites, with 5 and 10 wt% AC, were synthesized. The encapsulation was monitored by FT-IR and ^13^C-NMR. The presence of AC into the gel matrix has no significant effect on the thermal degradation of the composite but has a significant effect on the swelling degree, suggesting that the AC might interact with CDs, thus, acting as a further crosslinker. The performance of hydrogel composites towards pesticide removal was evaluated. Activated carbon has shown the best performance as adsorbent for IMD and CYM. It has also been shown that by increasing the amount of AC inside gel-matrix increases the RE and the maximum amount of sorbed pesticide. This cannot be solely explained by the modification of the BET surface area. Although those values are significant (higher than 56% and 80% RE for IMD-PCD/AC_5%_ and IMD-PCD/AC_10%_, respectively and maximum *q*_e_ of around 20 mg g^−1^ for both composites), they are lower than those obtained for the sorption of CYM and IMD onto AC—this work—but better than the majority of data reported for AC systems. Furthermore, the sorption mechanism follows the Sips model, suggesting that there is a dual (physical and chemical) interaction between the pesticides and the adsorbent. This is observed in the surface morphology of the adsorbent, in the absence and presence of pesticides, as well as it is in agreement with pseudo-second order sorption kinetics. It has been found that ionic strength (NaCl) and soil additive (urea) has no significant effect on the sorption isotherms of CYM and IMD, indicating that the interactions sorbent-sorbate are weak (e.g., London dispersion forces).

A relevant achievement of this work was to allow to conclude that these composites, unlike the AC, are able to sorb and desorb both pesticides, in significant amounts, after three cycles, without losing the ability to reach the maximum amount of sorbed/desorbed pesticide; in fact, it has been found that with the reuse cycles the maximum removal capacity of composites slightly increases. This accomplishment clearly paves the way for the development of new materials capable of acting simultaneously as sorbents and carriers, thus, contributing to the decrease of pesticides amounts applied in agriculture and, consequently, reducing the environmental damages.

## Data Availability

The data presented in this study are available within the article and in the Appendix A.

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
