# Peer review of "Poly(β-cyclodextrin)-Activated Carbon Gel Composites for Removal of Pesticides from Water"

_molecules, 2021, doi:10.3390/molecules26051426_

Round 1

Reviewer 1 Report

This paper deals with the design of PCD-activated carbon gel composites for removal of pesticides from water. The authors describe the synthesis of the composites, their charaterizations by several methods and their evaluation as sorbents for two model pesticides, CYM and IMD.  This paper is well documented and could deserve publication in Molecules after answering major concerns and the following comments.

  • Cyclodextrin compounds are known for their specific sorption properties over specific molecules (molecules showing specific affinity for cyclodextrin cavities). Surprisingly in this paper, the authors justify the use of a poly(cyclodextrin) matrix only because it has a low swelling degree. Many other hydrogel matrix when sufficiently crosslinked could play this role. Moreover the reader is expecting specific sorption properties when reading the title where polycyclodextrin is written as a first compound. The author should discuss on that aspect. I am expecting analysis with a third pesticide showing affinity for b-cyclodextrin and discussions on sorption behavior in this case.
  • Clearly activated carbon alone show the best results compared to the composite hydrogels. Why using a composite hydrogel? The author should justify that or conclude the AC are the best compounds.
  • Activated carbon compounds are not properly described. What are the sizes of the particles and what is the value of their specific surface? Fig 1c shows a schematic representation of the composite hydrogel and it looks like the AC particles are smaller than bCD cavities. Is it really the case? If not the schematic representation should be modified.
  • Lines 300-306: Authors say that sorption kinetics are comparable to PCD hydrogels. But the line after they conclude that AC is controlling the sorption. This seems to me in contradiction.
  • Lines 318-320. Why a slow kinetic shows that the process is based on hydrogen bond or inclusion complex chemisorption?
  • Line 384-386 The authors mention pesticide – pesticide interactions and their tendency to form oligomers. Do they have more direct argument to prove that, or references to cite?
  • Line 399: The authors mention RE and qm values in table 1. There should be a mistake.
  • Line 422-429. The authors discuss the effect of NaCl. They hypothesize CD /CD interactions that could be disrupted by screening effect of NaCl. However, these interactions could be screened only if the CD/CD interactions were of electrostatic nature. In the present case, I understood that PCD should be neutral. The authors should reconsider their interpretation.

How are the sorption/desorption cycles done? This is not described. In particular, how long does it take to desorb the same amount as adsorbed? How long do the author wait for the cycles described?

Author Response

Response to the reviewers’ comments:

Ms. ID.: molecules-1097813~

Title: Poly(β-cyclodextrin)-activated carbon gel composites for removal of pesticides from water

This paper deals with the design of PCD-activated carbon gel composites for removal of pesticides from water. The authors describe the synthesis of the composites, their characterizations by several methods and their evaluation as sorbents for two model pesticides, CYM and IMD. This paper is well documented and could deserve publication in Molecules after answering major concerns and the following comments.

  1. Clearly activated carbon alone show the best results compared to the composite hydrogels. Why using a composite hydrogel? The author should justify that or conclude the AC are the best compounds.

We thank reviewers to this issue although this point is clearly mentioned in the paper. The use of AC as adsorbent is clearly better than the composites; however the use of AC has some drawbacks (see lines 62-63). One of them is its re-use which is clearly highlighted in the Section 3.5. The conclusion that AC is the better adsorbent is already done in Conclusions (pages 466-467); however, we have decided to stress that by adding the following text in the Conclusions: “Activated carbon has shown the best performance as adsorbent for IMD and CYM.”

  1. Activated carbon compounds are not properly described. What are the sizes of the particles and what is the value of their specific surface? Fig 1c shows a schematic representation of the composite hydrogel and it looks like the AC particles are smaller than bCD cavities. Is it really the case? If not the schematic representation should be modified.

Activated carbon was received in cylindrical pellet-form and used without further treatment than manual shredding to obtain a thin powder as much as possible.

The Reviewer is absolutely right and Figure 1 has been modified accordingly. Besides the following text was added in the caption: “CD and AC structures are not at the same scale”

The surface area of and the average pore diameters for all samples are now reported in the Table 2.

  1. Lines 300-306: Authors say that sorption kinetics are comparable to PCD hydrogels. But the line after they conclude that AC is controlling the sorption. This seems to me in contradiction.

What we are trying to say is that long-time sorption process “ca. 10 and 20 hours” is mainly due to the low swelling degree of the PCD-matrix resulting in a slow sorption rate. However, even the presence of low amount of AC in the matrix gel has predominant effect on the sorption capability. To avoid misunderstandings between kinetics and isotherms the last phrase has been removed.

  1. Lines 318-320. Why a slow kinetic shows that the process is based on hydrogen bond or inclusion complex chemisorption?

The suggestion that a chemisorption mechanism occurs is mainly based on the occurrence of a PSO kinetics and not on the magnitude of k2 value; of course that once k2 changes with the adsorbate initial concentration also might suggest that a chemisorption process might occur.

The text has been modified to:

“Hence, a chemisorption nature is suggested which can be justified by the occurrence of hydrogen bonds or inclusion complexes formation.”

  1. Line 384-386 The authors mention pesticide – pesticide interactions and their tendency to form oligomers. Do they have more direct argument to prove that, or references to cite?

We really thank the Reviewer for this comments allowing a clarification. In fact, it is quite speculative to refer the formation of oligomers. Therefore, the phrase has been modified and now reads: “. Such a behaviour might be linked to the tendency of pesticides to form dimers [77], which might hampers the sorption process – as previously discussed.”

  1. Line 399: The authors mention RE and qm values in table 1. There should be a mistake.

Really sorry for that; in fact, there were two Tables 1. All tables have been checked and renumbered throughout the ms.

  1. Line 422-429. The authors discuss the effect of NaCl. They hypothesize CD /CD interactions that could be disrupted by screening effect of NaCl. However, these interactions could be screened only if the CD/CD interactions were of electrostatic nature. In the present case, I understood that PCD should be neutral. The authors should reconsider their interpretation.

We are really thankful to the Reviewer for raise this issue, which is not unanimous. In fact, the screening effect is usually related to interactions of electrostatic nature. However, H-bonding can be characterized as very strong dipole-dipole interactions; although these interactions are not of long-range interactions’ nature, the electrostatic nature is implicit and, consequently, the screening effect might occur. Two references were added to support our thoughts.

  1. How are the sorption/desorption cycles done? This is not described. In particular, how long does it take to desorb the same amount as adsorbed? How long do the author wait for the cycles described?

Each cycle takes 8 days long, 4 days for sorption and 4 days for desorption processes.

Reviewer 2 Report

The manuscript described the synthesis of composite material for the sorption and release of pesticides in soils. It is unclear to this reviewer how the topic fits “molecules” as the composite material is not a molecule. The chemical synthesis and characterization appears robust but several key points seem missing form the synthesis  description, making it unlikely that somebody can reproduce this. another  critical point is that the sorption tests are not correctly done (see below). Finally a big issue  the environmental relevancy of this. the context is provided as soil additive but the tests are performed in totally irrelevant conditions (no soil present, no real matrix and totally unrealistic conditions. While the scientific work might be ok the relevancy for that context (environmental application) is totally unclear.

Major Issues

The main goal is an environmental relevancy for use in soils? But how do any of the characterization experiments relate to this… e.g. a typical spray concentration is 1000ppm (Cohen, Y., and Grinberger, M. Control of metalaxyl-resistant causal agents of late blight in potato and tomato and downy mildew in cucumber by cymoxanil. Phytopathology 77(9), 1283-1288 (1987).) and you do experiments at 500ppm? So at spray concentration? In soil? Can you justify these concentrations? They are way too high for any environmental relevance. Also you do this in soil.. so one would expect some comparative sorption that your sorbent actually sorbs substantially more of the pesticide than the soil. Also any “real water matrix? For the vey least the experimental conditions would need to be discussed relative to a “real” environment and how they might or not be representative?

Sorption test: you cannot do a sorption test in a teabag! It is questionable that you mix well as the sorbent is kept together AND the bag itself might sorb pesticide to it. For the very least you need to show that this is not the case (e.g. sorption test with just the teabag etc)

The synthesis method and cleaning is incomplete and does not allow for easy replication e.g. stirring with NaOH? What does this mean? pH or amount  NaOH by amount of water should be given. On the other hand, the characterization is description is provided with many useless details like the pressure used to make the FT-IR pellets or the outer diameter of the zirconium NMR rotors?

More detailed issues:

Table1: many numbers are shown with excess digits, please only give statistically significant numbers

Table 1: Define RE clearer may be a exponent? To make people check bottom of the table

Table 1: Please write Latin names in italic

While overall the English is good, it could benefit form proofreading also names of chemicals etc. e.g. it is ACROS chemicals not ACROSS

Also the references in the text there should be a space between the word and the referen.

Some “editing:” signs in there (vertical lines next to the numbers)

Figures: some plots have error bars and many do not, please put error bars on all plots including the isotherms

Fig 6. Some of the surface structures look like possible artifacts because of drying process, can you comment on this?

Fig 6. The SME micrographs absolutely need to have a clearly visible and readable scale bar.

Fig 2c: OH peaks especially the 3400 peak… can you discuss how this relates to water in your sample..…

Author Response

Response to the reviewers’ comments:

Ms. ID.: molecules-1097813~

Title: Poly(β-cyclodextrin)-activated carbon gel composites for removal of pesticides from water

  1. The main goal is an environmental relevancy for use in soils? But how do any of the characterization experiments relate to this… e.g. a typical spray concentration is 1000ppm (Cohen, Y., and Grinberger, M. Control of metalaxyl-resistant causal agents of late blight in potato and tomato and downy mildew in cucumber by cymoxanil. Phytopathology 77(9), 1283-1288 (1987).) and you do experiments at 500ppm? So, at spray concentration? In soil? Can you justify these concentrations?

IMD and CYM have maximum water solubility of 600 mg mL-1, so we performed kinetics and isotherms sorption until the maximum. Usually, the spray formulation are not solution but suspension where the pesticides are not completely solubilized. Moreover, considering the aqueous media, the maximum residual levels are 0.01 mg/kg in Europe when other specification are not defined. Moreover we understand that in the mentioned reference, they need to reproduce the conditions of pesticide application to determine the effect of the cymoxanil on late blight. That is not our case or purpose.

  1. They are way too high for any environmental relevance. Also you do this in soil.. so one would expect some comparative sorption that your sorbent actually sorbs substantially more of the pesticide than the soil. Also any “real water matrix? For the vey least the experimental conditions would need to be discussed relative to a “real” environment and how they might or not be representative?

We think that there is a misunderstanding. The study was performed in aqueous media. In that preliminary study, we assessed the material performance for pesticides removal before any study on real water matrix. At this regard, we used the maximum range of concentration in order to understand the material capability with the increase of the pesticide concentration. Surprisingly, the adsorbent materials present a higher RE% at lower IMD and CYM concentration, in particular the composite materials (Fig. 7).

  1. Sorption test: you cannot do a sorption test in a teabag! It is questionable that you mix well as the sorbent is kept together AND the bag itself might sorb pesticide to it. For the very least you need to show that this is not the case (e.g. sorption test with just the teabag etc).

We disagree with the reviewer; in fact, the use of a teabag is not original and all tests to ensure that teabags are neutral in the sorption process have been carried out. Besides, teabags were used exactly to ensure that all adsorbent were immersed in the solution. A statement indicating that no measurable sorption onto teabags has been detected were added: “Teabags were used to ensure a complete immersion of the adsorbents in the bulk phase and preliminary experiments have shown that no measurable sorption of pesticides onto teabags took place.”

  1. The synthesis method and cleaning is incomplete and does not allow for easy replication e.g. stirring with NaOH? What does this mean? pH or amount NaOH by amount of water should be given.

This is probably a misunderstood. We only have used NaOH (solid) in the DMF drying process. There is no water involved in that process. In any case the word “(solid)” has been added after NaOH.

  1. On the other hand, the characterization is description is provided with many useless details like the pressure used to make the FT-IR pellets or the outer diameter of the zirconium NMR rotors?

We have followed Reviewer’s suggestion and the description of techniques has been shortened

More detailed issues:

  1. Table1: many numbers are shown with excess digits, please only give statistically significant numbers

The values reported in Table 1 were obtained from literature and, consequently, we must be accurate to cite the values obtained by our peers. In any case, we have decrease the significant numbers in some cases.

  1. Table 1: Define RE clearer may be a exponent? To make people check bottom of the table

We have followed the reviewer suggestion and a note has been added at the bottom of Table.

  1. Table 1: Please write Latin names in italic

The text has been modified accordingly.

  1. While overall the English is good, it could benefit form proofreading also names of chemicals etc. e.g. it is ACROS chemicals not ACROSS

We have read the ms and checked the language.

  1. Also the references in the text there should be a space between the word and the referen.

We went throughout the ms and all references have been checked and the ms was modified accordingly.

  1. Some “editing:” signs in there (vertical lines next to the numbers)

We have checked the ms and unfortunately we were not able to see such “vertical lines”.

  1. Figures: some plots have error bars and many do not, please put error bars on all plots including the isotherms

All plots have error bars, however, especially on the sorption isotherms, the error is smaller than the data points and, consequently, error bars are not seen.

  1. Fig 6. Some of the surface structures look like possible artifacts because of drying process, can you comment on this?

The SEM analysis can be tricky due to the occurrence of the artifacts as suggested by the Reviewer. We follow a very strict protocol to observe the surface morphology as real as possible (i.e., before drying); besides the PCD-based gels do not swell too much, consequently, the drying process does not affect too much the surface morphology as in the case of superabsorbents gels. We got other micrographs and the chosen ones are, from our point of view, well representative.

  1. Fig 6. The SME micrographs absolutely need to have a clearly visible and readable scale bar.

The figure has been changed to accommodate this suggestion

  1. Fig 2c: OH peaks especially the 3400 peak… can you discuss how this relates to water in your sample..…

We are quite sure that the peak at 3400 cm-1 related to -OH groups is not related to water presence in the samples because they are freeze-dried and stored on a drier with silica, although some bound (so-called non-freezing) water can remain in the samples. Additionally, CDs can present intermolecular hydrogen bonds that contribute to the broadening of the peak.

Reviewer 3 Report

The subject matter of submitted manuscript is included in the subjects of the Molecules. The title does adequately describe the contents of the paper and the abstract is informative enough. The subject matter is very interesting, since the removal of pollutants, especially pesticides, from environment in very important. The experimental part is well designed, Authors used a lot of techniques for the characterization of formed solids and the discussion is adequately supported from the experimental data. The submitted manuscript in the present form is suitable for publication in the Molecules and it will be further improved after some additions/corrections.

Some additional comments:

  • What was the mean yield of poly(β-C) synthesis?
  • What were the specific surface area of prepared materials?
  • What was the number of repetitions in sorption experiments?

Author Response

Response to the reviewers’ comments:

Ms. ID.: molecules-1097813~

Title: Poly(β-cyclodextrin)-activated carbon gel composites for removal of pesticides from water

  1. What was the mean yield of poly(β-C) synthesis?

The mean percentage yields by weight are: PCD=89.3%; PCD-AC5%=92.5%; PCD-AC10%=78.0%. This information has been in the ms (Lines 124-125)

  1. What were the specific surface area of prepared materials?

This information is now in the Table 2.

  1. What was the number of repetitions in sorption experiments?

All experiments were carried out in triplicate. This is now described in lines 218 and 226.

Reviewer 4 Report

This paper is interesting and sufficiently innovative. However, there isn't some important information: the pcz of the material and its surface area. Moreover, as far as literature is concerned, we lack some important references relating to the use of mesoporous materials such as for example

"Decontamination of waters polluted with simazine by sorption on mesoporous metal oxides” Journal of Hazardous Materials (2011) vol. 196, pp. 242-247, and "Removal of agrochemicals from water by adsorption: a critical comparison among humic-like substances, zeolites, porous oxides and magnetic nanocomposites" Processes 2020, 8, doi:10.3390/pr8020141, pp. 1-26 and references contained therein.

A moderate english language is required

Author Response

Response to the reviewers’ comments:

Ms. ID.: molecules-1097813~

Title: Poly(β-cyclodextrin)-activated carbon gel composites for removal of pesticides from water

  1. This paper is interesting and sufficiently innovative. However, there isn't some important information: the pcz of the material and its surface area.

The table 2 has been updated with surface are and average pore diameter for all materials. Concerning pcz we think that is not useful once all materials are neutral

  1. Moreover, as far as literature is concerned, we lack some important references relating to the use of mesoporous materials such as for exemple…

All references suggested by the reviewer were added in the Introduction section (new refs. 18 and 21).

  1. A moderate english language is required

We did our best to improve the language.

Round 2

Reviewer 2 Report

If the point is that the materials take up pesticides from the environment, the concentrations need to make sense.. they do not! the concentrations used here are way higher than anything that is observed anywhere in the environment. Or frame it different and do not talk about removal from water and soil... or show actual measurements that justify why you work at these concentrations.

the teabags might not have any effect in your experiments as the concentartiosn are so high, still data would help not just writing that they did not have an effect.

Author Response

If the point is that the materials take up pesticides from the environment, the concentrations need tomake sense. they do not! the concentrations used here are way higher than anything that is observed anywhere in the environment. Or frame it different and do not talk about removal from water and soil... or show actual measurements that justify why you work at these concentrations.

In the 1st report, the Reviewer#2 wrote “ ...e.g. a typical spray concentration is 1000ppm (Cohen, Y., and Grinberger, M. Control of metalaxyl-resistant causal agents of late blight in potato and tomato and downy mildew in cucumber by cymoxanil. Phytopathology 77(9), 1283-1288 (1987).) and you do experiments at 500ppm?.”

Once the reviewer has apparently no argument to contest our reply, he/she wrote in the 2nd report “...the concentrations used here are way higher than anything that is observed anywhere in the environment...”

The authors do not understand the scope, nor the purpose, of these contradictory comments and request.

the teabags might not have any effect in your experiments as the concentartiosn are so high, still data would help not just writing that they did not have an effect.

The text has been changed to (Lines 176-179): “In the concentration range at which pesticide sorptionwas evaluated, control experiments were carried out showing that no measurable sorption of pesticides onto teabags has been detected.”. Additionally, the teabag is described in terms of material and mesh size.

A list of papers where teabags are used to ensure the complete immersion of low density materials are provided:

http://dx.doi.org/10.3390/molecules25204628

http://dx.doi.org/10.1039/c8ra02332h

http://dx.doi.org/10.1016/j.proche.2012.06.038

http://dx.doi.org/10.1016/j.polymer.2005.04.015

http://dx.doi.org/10.1016/j.colsurfa.2017.02.018

http://dx.doi.org/10.1039/C8TA02680G

http://dx.doi.org/10.1021/acsomega.0c02634

Reviewer 4 Report

The manuscript can now be accepted in the present form

Author Response

Thank you for positive comments.